# The Use of Sacrificial Graphite-like Coating to Improve Fusion Efficiency of Copper in Selective Laser Melting

**DOI:** 10.3390/ma16062460

**Published:** 2023-03-20

**Authors:** Angela Elisa Crespi, Guillaume Nordet, Patrice Peyre, Charles Ballage, Marie-Christine Hugon, Patrick Chapon, Tiberiu Minea

**Affiliations:** 1Laboratoire de Physique des Gaz et Plasmas, LPGP, Université Paris-Saclay, CNRS, F-91405 Orsay Cedex, Francetiberiu.minea@universite-paris-saclay.fr (T.M.); 2Groupe de Recherches sur l’Energétique de Milieux Ionisé GREMI, Université d’Orléans, CNRS UMR7344 14 Rue d’Issoudun BP6744, 45067 Orléans Cedex 2, France; 3Procédés et Ingénierie en Mécanique et Matériaux, PIMM, Hesam, CNRS Cnam, Arts et Métiers Sciences et Tecnologies, 151 Bd de l’Hôpital, 75013 Paris, France; 4HORIBA Scientific 14 Boulevard Thomas Gobert, Pass. Jobin-Yvon, 91120 Palaiseau, France

**Keywords:** selective laser melting, copper, amorphous carbon, sacrificial layer, interferences, power density

## Abstract

Thin and ultrathin carbon films reduce the laser energy required for copper powder fusion in selective laser melting (SLM). The low absorption of infrared (IR) radiation and its excellent thermal conductivity leads to an intricate combination of processing parameters to obtain high-quality printed parts in SLM. Two carbon-based sacrificial thin films were deposited onto copper to facilitate light absorption into the copper substrates. Graphite-like (3.5 µm) and ultra-thin (25 nm) amorphous carbon films were deposited by aerosol spraying and direct current magnetron sputtering, respectively. The melting was analyzed for several IR (1.06 µm) laser powers in order to observe the coating influence on the energy absorption. Scanning electron microscopy showed the topography and cross-section of the thermally affected area, electron backscatter diffraction provided the surface chemical composition of the films, and glow-discharge optical emission spectroscopy (GDOES) allowed the tracking of the in-deep chemical composition of the 3D printed parts using carbon film-covered copper. Ultra-thin films of a few tens of nanometers could reduce fusion energy by about 40%, enhanced by interferences phenomena. Despite the lower energy required, the melting maintained good quality and high wettability when using top carbon coatings. A copper part was SLM printed and associated with 25 nm of carbon deposition between two copper layers. The chemical composition analysis demonstrated that the carbon was intrinsically removed during the fusion process, preserving the high purity of the copper part.

## 1. Introduction

Copper fusion is challenging because its high thermal conductivity results in important local temperature gradients and fast solidification during Cu-selective laser melting (SLM). In addition, Cu is one of the most reflective metals in the IR range, with 99% reflectivity [1,2,3]. SLM is a well-known process that uses a laser beam to melt metallic powder bed in layers to form a 3D object. In the case of Cu, it challenges the process considering it is a hard metal to melt. Some issues are poor absorptivity, denudation (powder spattering), intermittent melting or balling effect, and laser penetration. The estimation of the laser absorption for metals in SLM is a difficult task [4]. The absorptivity is connected to the laser penetration and melting pool depth. The Cu solid-state absorption is around 2%. It can drastically increase in a liquid state (melting), mainly around the green wavelength [5,6]. Hence, Cu fusion requires different energy management of the laser-surface interface to obtain an acceptable printed object [7,8]. Recent works have achieved pure Cu pieces printed using an IR laser employing an energy density of 103 kW/mm^2^. Using copper alloys such as Cu-Cr-Zr-Ti and Cu-Al-Ni also attained quality objects employing an energy density of 640 kW/mm^2^ [9]. The majority of recent work reports hundred of kW/mm^2^ for pure and Cu alloys [9,10,11].

Several strategies with carbon are used to reduce copper fusion energy. Specifically, carbon has been added to copper in differrent forms, such as nanoparticles or carbon coatings, all around the Cu powder grains, using ex situ processes [12]. Generally, carbon addition to copper leads to better laser absorption and less laser power required for good-quality 3D printing [12,13,14]. The main drawback when adding carbon to copper is its segregation generating pores and cracks in the final metal part [15,16,17]. Preliminary studies have shown reduced copper reflectance when using polymeric materials as an absorbing layer [13,18]. In addition, using a multilayer stack of several graded metal-carbon layers followed by an amorphous carbon film produces excellent selective solar absorbers [19]. Nevertheless, pure copper’s physical properties are significantly affected by impurities [20]. Thus, with the recent development of printable graphene conductor inks, the joint of a graphite-like layer in minimal quantities that does not alter Cu properties appears to be an easy alternative to manage Cu’s fusion using IR energy [15,21]. However, the use and effects of thin and ultrathin layers as absorbers or anti-reflectives in the additive manufacturing processes as SLM still lack studies.

The main objective of the present work is to propose an efficient dark thin film using carbon to enhance IR light absorption into Cu and hence induce an effective local fusion without harming Cu resistivity. Also, the goal is to reduce carbon addition and maximize the absorption without residues after fusion [22]. Here the focus goes beyond analyzing the behavior of carbon when covering only the top-most surface of the powder under SLM.

## 2. Methodology

Measurements have been performed on a bench equipped with an IR Trumpf Laser (TruDisk 10002, Trumpf Laser GmbH, Schramberg, Germany, *λ* = 1.03 µm) with a top-hat beam-shaped beam. Laser power varied between 600–2000 W. The laser was focused on the samples and moved at a constant speed under Ar protective atmosphere. The laser bench mime single track of the SLM process. The main laser parameters are summarized in Table 1. The surface power density was calculated using Equation (1).
(1)$$\Phi =\frac{4P}{\pi D^{2}}$$
where *Φ* is the power surface density (W/mm^2^), P is the laser power (W), and D is the spot diameter (mm) of the focused laser on the sample.

The first set of samples consisted of a graphite-like aerosol (commercial aerosol containing 5% wt. graphite—GRAPHIT 33) deposited on a pure Cu powder and its native oxides. The powder size distribution was 45–106 µm. The spray distributed the aerosol from about 30 cm above the powder leading to a thin non-uniform coating of 3.5 ± 1.5 μm. The topography and cross-sections are analyzed. The cross-sections followed the standard metallography procedure for Cu. Then the electropolishing used a Struers LectroPol 5 (Struers Inc., Champigny-sur-Marne, France) with a D1 commercial solution for 12 s [23]. The samples are referred to as PCu-i and C-i, followed by the laser power (i) used in the fusion for pure and coated samples, respectively. The laser powder sequence used was: 600 W, 800 W, 1000 W, 1200 W, 1500 W, and 2000 W. The highest energy used was 2000 W, which was used as a reference to the energy reduction calculations in this work.

The second batch of samples used optimized ultra-thin films, inspired by [24], in the range where interference on copper is observed to validate the hypothesis that the carbon is removed from the molten track if the minimal thickness is used.

Analyzing the reflectance of the carbon-coated copper (Figure 1a), when depositing is about 70 nm thick, the reflection is reduced to around 30% for IR (~1 µm wavelength). However, the deposition of such a layer is time-consuming, and there is a risk of adding C (traces) in the bulk Cu material. When depositing 25 nm, the reflectance is reduced to around 50%.

Therefore, the choice of thickness was guided by the highest variation of the reflectance issued from its first derivative versus the thickness of the coating. Indeed, in this way, the deposition time and contamination probability are minimized. Also, for the minimum C added, the effect on the reflectance reduction is the highest, as demonstrated by both theoretical and experimental results [22]. From Figure 1, the highest slope in reflectance (panel a) is observed for the minimum point (highest absolute value) of the first derivative (panel b). The derivation of the experimental and theoretical data of amorphous carbon film (a-C) grown by direct current magnetron sputtering (DCMS) on 110 µm thick Cu foils (99.999% Cu purity- FCu-i) confirms that a 50% reduction within the smallest thickness is around 25 ± 5 nm of the pure amorphous carbon film. This batch is called a-C25-(i) 600 W and 1200 W where i = 600 W, 800 W, 1000 W, 1200 W, and 2000 W, respectively. The as-deposited a-C is called the ‘Reference a-C’ film. Details about the a-C can be found elsewhere [22].

Preliminary tests were done for a 3D-printed object by SLM. Several small squares of 2.5 mm × 2.5 mm were printed at each layer (~60 µm thick), producing small cubes, using a conventional Cu building plate and thin powder (15–45 µm). When the cubes reached a few millimeters in height, the process was stopped, and the plate was removed from the 3D printer. A total of 25 nm of a-C film was deposited ex situ using a physical vapor deposition reactor. The reactor is a multi-target equipped with several 6-inch diameter magnetrons (Alcatel, Boulogne-Billancourt, France). It is composed of a vacuum chamber exhausted by a molecular turbo pump backed by a primary pump. A target in graphite was used with the same conditions as the a-C films deposited onto the foils and powder [22]. A load-lock chamber was employed to transfer the building plate and the cubes using a movable sample holder under the carbon target. After the a-C deposition, the building plate was introduced again into the SLM machine. The printing process continued fusing around 300 µm (5 layers of powder) over the a-C deposition using around 1000 W of laser power. This sample combined the SLM and DCMS processes. It is referred to as Printed-part-Cu/a-C.

All samples presented and discussed in this study are summarized in Table 2.

Energy Dispersive X-ray Spectroscopy (EDS) chemical analyses for the 25 nm samples were carried out using a ZEISS Sigma HD Scanning Electron Microscope (SEM) (ZEISS, Jena, Germany). This microscope operates in backscattering mode, equipped with an electron backscatter diffraction (EBSD) system and a NORDIF CD camera. The software for quantifying the crystallography grains’ orientation was OIM™ version 8.5. Optical images were taken with a Zeiss Axio imager M2. Melting pool penetration depth employed the software Image J version 1.53t. An SA2 Photron–Fastcam high-speed camera recorded the experiments’ evolution to investigate the melt-pool behavior and eventual spatter ejections. Reflectance was determined with an integrating sphere located 3 mm before the focal length of the IR Trumpf laser (Trumpf Laser GmbH, Schramberg, Germany). The laser power utilized in the reflectance measurements was 60 W. The composition in-deep was analyzed with a Glow Discharge Optical Emission Spectrometry (GDOES) GD-Profiler 2™. For the measurements, the building plate and three cubes were cut using electroplate diamond griding. The printing plate thickness was removed, and the cubes were homogeneously compressed to reduce their total height. The initial height was about 1 mm reduce by half of their thickness, 0.5 mm.

## 3. Results and Discussions

### 3.1. Topography Analysis

The evolution of the fusion topography for pure and coated Cu is demonstrated in Figure 2a,b. In the sequence of Figure 2a, only intermittent melting or balling effect is observed until 1500 W. At PCu-1500, the periodic melting starts to touch, indicating the beginning of a continuous fusion. PCu-2000 showed a more evident fusion track. As the energy increased, the melted copper changed from spherical to elliptical Cu drops [25]. In C-1200, the melted Cu became more elongated [26]. A clear difference struck the eye when Cu was coated with a graphite-like film. Much more powder was fused to the plate with a few discontinuities. Figure 2b demonstrates the analogous samples coated with graphite-like spray. For the C-600, one can observe some intermittent melting. However, the track is generally continuous for higher energy, such as C-1200, C-1500, and C-2000 [25].

For the pure Cu series, the previous uniform powder bed left the sides of the track, the so-called denudation. The denudation effect happens when the Cu powder flies and spatters due to laser pressure or reflectance, mainly when the processing energy is insufficient to melt Cu, resulting in a lack of powder around the fusion [27]. When the graphite is present, the denudation effect is reduced for laser energies above 1000 W. The spatter reduction between balling and continuous-stable fusion has already been observed. The denudation reduction is important during the transition spatter/stable fusion and corroborates the penetration depth of the laser. For higher powers resulting in deeper penetration, the denudation effect is directly linked to a too-deep melting pool with the keyhole as vaporization [28]. In this context, when the graphite layer is over the powder, the energy is sufficiently absorbed to melt a continuous track resulting in a minimal denudation effect [29,30].

The evolution of the Cu fusion is observed, from a few drops to an almost continuous track in Figure 3a–f. As the energy increases, the round balling becomes larger and elongated drops of around 10 µm. The extended drops are generated by high scan speed due to incomplete wetting and spreading of molten droplets observed in the left column [31]. The shape of the balling indicates the melting pool shape during fusion. The Cu solidification is faster around its boiling temperature, resulting in more significant balling tendencies than titanium. Surface instability of single scan tracks, like balling and irregular track width, can lead to the formation of pores and deteriorate the sample quality [32]. The SEM images confirm that the fusion drastically improves when the graphite spray is on the sample (right column) [33]. The presence of graphite reduces the energy needed for a continuous fusion to 1200 W (38 kW/mm^2^). Higher energy input improves the wetting and the quality of fusion. On the contrary, an almost continuous fusion is obtained at 2000 W (64 kW/mm^2^) for pure Cu [34]. Considering that 1200 W is enough to get a continuous fusion when carbon is present, the reduction in the power laser reaches 40% to fuse copper in this experimental series. When it comes to the literature comparison, this reduction can be much more expressive since densities such as 100 kW/mm^2^ are rarely obtained for Cu or Cu alloys [18,35].

The effect of the 25 nm a-C on copper foils is demonstrated in Figure 4a–e. Pure copper foils (left column) did not reveal the fusion track position. The foils showed minor melting at the edges and minimal oxidation in the laser track for higher, for 2000 W Figure 4e. On the contrary, all the fusion can be observed in the presence of a-C films (right column). The track thickness increases with the laser power. The arrows indicate where the foil was cut using higher laser power Figure 4d,e. Using a copper foil, one can observe the overuse of laser power discussed later in Section 3.3.

### 3.2. Cross-Section Analysis

To check the quality of the fusion joints, we analyzed the cross-sections of the fused powder using graphite films.

In Figure 5a, one can observe the cross-sections for pure Cu powder in several laser powers. The cross-section analysis for energies, such as 600 W, was not possible. The metallography preparation may severely affect the sample at such low energies, indicating the poor quality of the fusion. The first visible cross-section is observed at 1000 W. When increasing the power to 1200 W (Figure 4b), the center of the bead after fusion is correctly welded to the plate (green dotted line), but the edges are still not connected (red lines in Figure 5d). The same is observed when using 2000 W (~64 W/mm^2^) (Figure 5c,e). Although a bigger area in the center is well-fused, failures remain in the fusion edges (Figure 5e) [7,10,11].

The graphite-like coated sample cross-section analysis shows samples C-600 and C-1200. Figure 6a shows that a quality fusion was obtained using only 600 W around 19 kW/mm^2^. In the upper part of the weld, some pores are detected. The number of pores decreases with the laser power increase. The cross-section shows typical grains for Cu structure. The microstructure changes along the vertical axis of the bead due to thermal gradients. The elongated grains in the base turn into classical equiaxial in the upper part because of rapid solidification [36]. Globally, the shape of C-1200 is more regular (Figure 5b) than PCu-1200 (Figure 6b). The C-1200 is more prone to provide better overlap for successive fused layers during SLM. Carbon does not emerge in the grain boundaries [37].

The inverse pole figure in Figure 4c,d clears out the grain junction in the region between the fused powder and the building plate, the transition zone. A great welded joint at the transition zone without cracks or pores is observed, even for the lower power case, C-600. Despite good fusion, the deepest part of the molten pool is around 12 ± 0.5 and 22 ± 0.7 μm in the C-600 and C-1200, respectively. The bottom part of the fused beat is flatter than the conventional pure Cu in SLM [38]. Such a low power density as 19 kW/mm^2^ (C-600) was not found in the literature for SLM involving IR laser with Cu fusion [39]. Generally, energies in the order of hundreds of kW/mm^2^ are involved in the process [10,40,41].

The coating method used in this work covers only the upper part of the grains, reducing the deposited carbon quantity in a single track. Pure Cu characteristics are preserved in the contact region with the underlayer since the graphite-like is only deposited on the surface exposed to the laser. This deposition produces graphite hollow hemispheres of 3.5 µm wall. The continuous hollow hemispheres result in 6.5 × 10^−6^% wt of graphite over all the Cu beads of 55 µm (the distribution mode) for the track volume.

### 3.3. Composition Analysis Using Ultrathin Carbon Films

The effect of a minimal carbon film on copper is enhanced due to interference phenomena, as analyzed and explained for an ultra-thin amorphous carbon coating of Cu [22]. Previous studies demonstrate that larger thicknesses of carbon, such as the one deposited with the aerosol, may be less effective than ultra-thin films [24]. In Figure 7a, a representative EDS spectrum of several points of measurement shows the composition of the sample. In Figure 7b,c, SEM images show the measurement position. The EDS of a-C25-600 (green line in Figure 7a) reveals that thermal processes considerably remove a-C from the fusion track compared to the Reference-a-C-film. The relative intensity of the carbon peak decreases significantly while the *Cu* increases (counts *y*-axis). The Cu peak reaches its highest value for a-C25-1200 if correlated to the relative intensity measured for the Reference-a-C-film and the fused sample using 600 W. The EDS analysis for pure Cu is similar to the one obtained for a-C25-1200; a negligible carbon peak is always present. Carbon can also come either from the EDS chamber or an organic form. Contamination could not be ruled out.

a-C coating absorbs more photons and evaporates together with the underlying copper. On the left side of Figure 7c, Cu foil is cut during the fusion. The cutting confirms that for 1200 W, the laser can remove up to 110 μm of Cu (foil thickness). As mentioned in the introduction, Cu reflects about 99% of the IR range spectrum, challenging its processing with light, such as SLM or even laser cutting. The integrating sphere measurements demonstrate that the thin a-C layer improves absorption when it covers the Cu substrates. Only 25 nm a-C absorbs around 50% more than pure Cu, as discussed in the methodology section. Consequently, less energy is lost by reflection utilizing ultra-thin films. Previous optical analysis of the same type of film deposited by magnetron sputtering using a spectrometer UV-Vis confirms these results [22].

It is common to boost the laser power in SLM to overcome Cu features, as mentioned in Section 3.1 [7,38]. Excess energy induces the evaporation of a significant quantity of Cu, increasing the local recoil pressure. Once the metal is melted, the laser absorption coefficient often increases compared to the solid state resulting in clouds of vapor metal above the molten pool [37]. The vapor expansion and superheated gas generate a depression in the molten pool, trapping laser energy and further enhancing absorption, forming a too-depth molten pool: the keyhole. This is undesirable because it degrades the printed object quality [42,43]. For C-1200, the recoil pressure makes the Cu foil bend since they were not fixed to the plate during fusion, as indicated in Figure 8a,b.

The striations and the holes in the foil confirm that sample C25-1200 attains the necessary energy to cut the copper foil and create significant recoil pressure [24]. The difference in the topography images of pure and coated Cu foils indicates that the energy is lost. The necessary power to observe the same effect on pure copper is hundreds of kW/mm^2^, much higher than the one used in coated samples [44]. In addition, the compromise between the final quality of the parts and avoiding the keyhole regimen is difficult to find for pure Cu. The details of this mechanism still need to be clarified [45].

The deposition of 25 nm of a-C before fusion as a top layer on copper represents about 0.004% wt. supposing that the laser does not remove the film during the fusion process. Indeed, the Reference-a-C-film shows a small carbon peak, significantly reduced after the fusion (1200 W). The results show that the carbon concentration in the final Cu part is minimal compared to its initial concentration. To confirm this finding, cubes printed by SLM were analyzed by GD-OES. The cubes combine DCMS deposition and SLM process. The three steps of sandwich Cu/a-C/Cu are depicted in Figure 9a–c. In Figure 9a, one can see the basis of the small cubes printed by SLM with pure Cu; (b) the surface of the plate surmounted by the small cubes covered with an ultra-thin film of a-C; and (c) the final cubes when Cu layers were printed on the a-C film.

One can observe that the characteristic copper color observed in Figure 8a turns into darker tons in Figure 8b. The difference in the colors of the a-C film suggests that the deposition is not perfectly homogenous in the edges of the printed part (less than 25 nm deposition). The blue tones confirm that most of the surface received the carbon coating, and interferences can occur, reproducing the color observed for other substrates [22]. In Figure 9c, one can see that the fused parts (square basis) continue to grow with the specific Cu color. The layers adhered well to the previous printing. The color of the non-printed area preserved the blue tons. Considering these results, a good strategy would be to deposit carbon only in the laser track so it would simply be removed, facilitating powder recycling [46].

The GDOES results are displayed in Figure 10a,b. The elements of interest, Cu, C, and O, were followed along the depth profile for the compressed sample surface, for which we are sure that the carbon layer was intercepted during the crater formation. The schematic cross-section of the analyzed region is shown in Figure 10b. The topography craters after compression are shown in Figure 10c.

The representative analysis of GDOES did not show any residue of the a-C film in bulk, as indicated in Figure 10a. One can see a higher quantity of carbon and oxygen on the surface. This explains the presence of carbon in the EDS surface signal (Figure 7). The copper signal is constant throughout the depth analysis, and C and O decrease. However, it is crucial to note the absence of any local increase in C concentration that could originate from C traces embedded into the Cu matrix.

The laser’s excess energy can melt and evaporate copper, removing carbon simultaneously [12]. One can speculate that by increasing the absorption of the laser power by reducing Cu reflectance, the graphite-like film has sufficient energy to bond with residual oxygen and hydrogen. When carbon nanoparticles are added to the powder, the amount of carbon and oxygen reduces after fusion compared to the corresponding powder before the printing process [12,24]. Vapors containing carbon can be found for temperatures exceeding 2400 °C, while Cu evaporates at 2595 °C. Thus, it leaves the fusion zone in any stable form of gas, such as CO_2,_ or driven by Cu evaporation, a current problem in SLM for Ti_6_Al_4_V and copper [22,38,47]. These vapors condensate, and some may cause damage to the laser optics [48]. In the copper case, the use of boosted laser energy is also problematic for the lens lifetime, pushing the development of this component [49]. However, the precise way carbon leaves the laser track on SLM needs further studies.

## 4. Conclusions

Suitable Cu powder fusion using an IR laser is obtained by reducing 70% of the laser power when depositing a sacrificial graphite-like layer over Cu substrates. The cross-section of the beads reveals elongated columnar grains in the base, near the building plate, turning in equiaxial on the top due to thermal gradients. The fusion transition between the powder and building plate zone did not show segregation cracks or pores. Thus, the deposition coating on the top of the powder preserves the pure Cu in contact with the previous (fused) layer composing the part. The deposition of the powder on only one side is essential for the energy transfer to previous layers. The better wetting angle for multitrack SLM is observed when using 1200 W. This results in a reduction of 40% in the IR laser power, considering the highest energy used in this work, around 2000 W.

Further, the sacrificial layer method is efficient since GDOES and EDS analysis observed negligible carbon residue after fusion for ultra-thin a-C film on copper foils and a 3D-printed object. The results show that interferences can increase the energy transfer reducing losses in reflected laser energy to copper using an ultra-thin graphite-like layer. This application can be extended to other energy absorption purposes like lightweight, flexible devices. The thin film deposition in this work is scalable to a large area sample.

## Figures and Tables

**Figure 1 materials-16-02460-f001:**
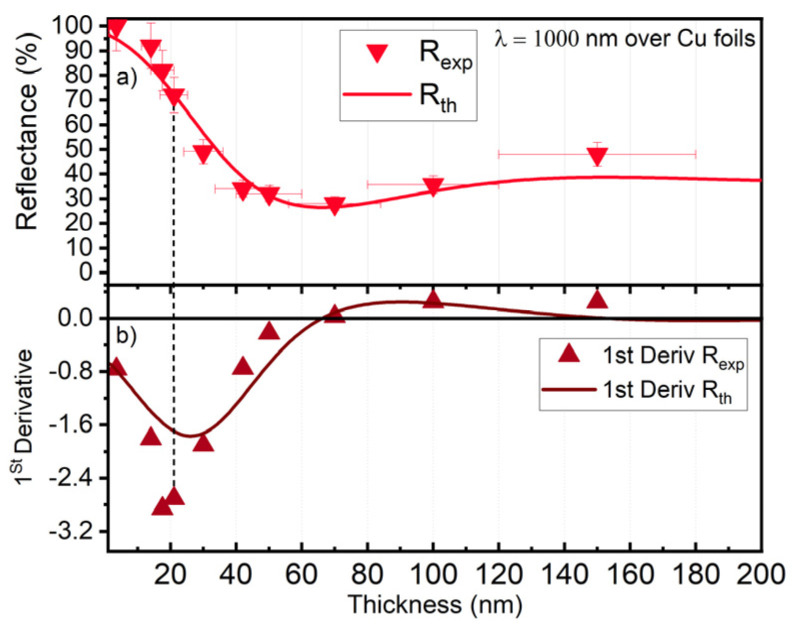
(**a**) Cu foils covered with a-C films xperimental (R_exp_) and theoretical (R_th_) reflectance for versus the thickness of a-C films; (**b**) The first derivative of results (**a**). The dashed lines indicate the minimum a-C thickness for the maximum reflectance reduction around 25 ± 5 nm.

**Figure 2 materials-16-02460-f002:**
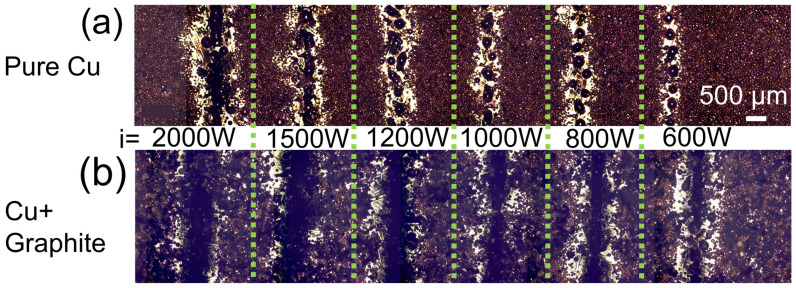
Optical images of the powder bed of after fusion using several laser powers. (**a**) Pure Cu powder (PCu-i), and (**b**) Cu powder covered with graphite spray (C-i) where i is indicated in the image. The scale in (**a**) is common for (**b**).

**Figure 3 materials-16-02460-f003:**
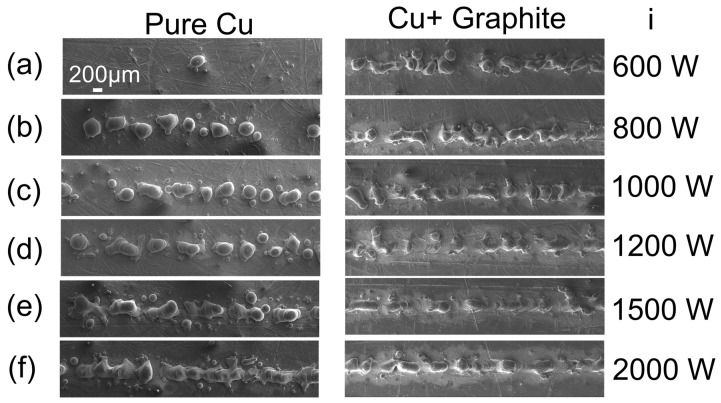
Topography SEM images of pure copper in the **left** column (pCu-i), and covered with graphite-like film (**right**/C-i) using the laser power of (**a**) 600 W; (**b**) 800 W, (**c**) 1000 W (**d**) 1200 W (**e**) 1500 W (**f**) 2000 W. The scale display in (**a**) is common for all the images.

**Figure 4 materials-16-02460-f004:**
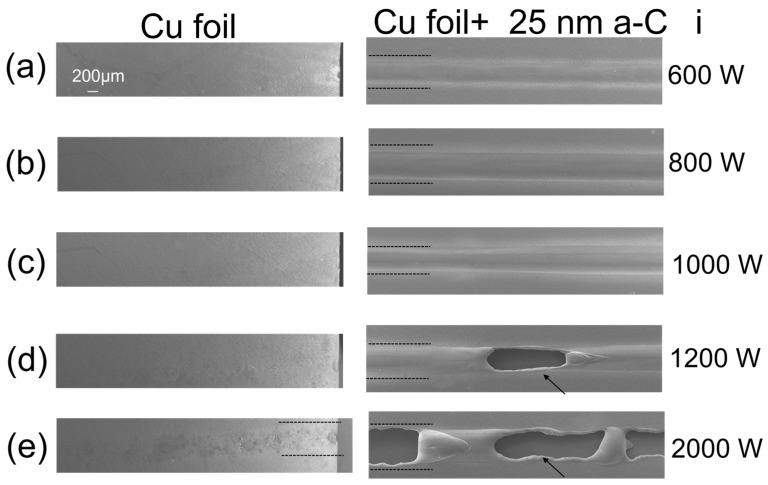
Topography SEM images of Cu foil (FCu-i/**right**) and covered with a-C films (a-C25/**left**) using the laser power of (**a**) 600 W; (**b**) 800 W (**c**) 1000 W (**d**) 1200 W and (**e**) 2000 W.

**Figure 5 materials-16-02460-f005:**
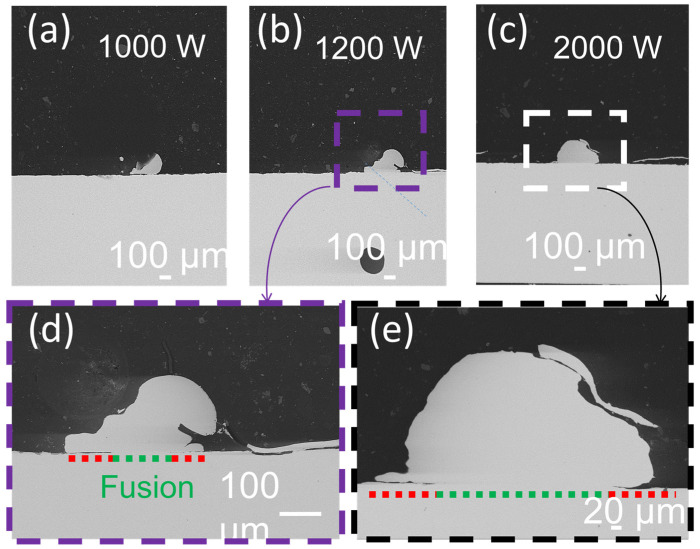
Pure copper cross-section welded bead SEM backscattered images for different laser power. (**a**) 1000 W, (**b**) 1200 W, (**c**) 2000 W. Zoom for (**d**) 1200 W and (**e**) 2000 W. The dotted lines indicate the fused zones in green and the not completely fused to the plate in red.

**Figure 6 materials-16-02460-f006:**
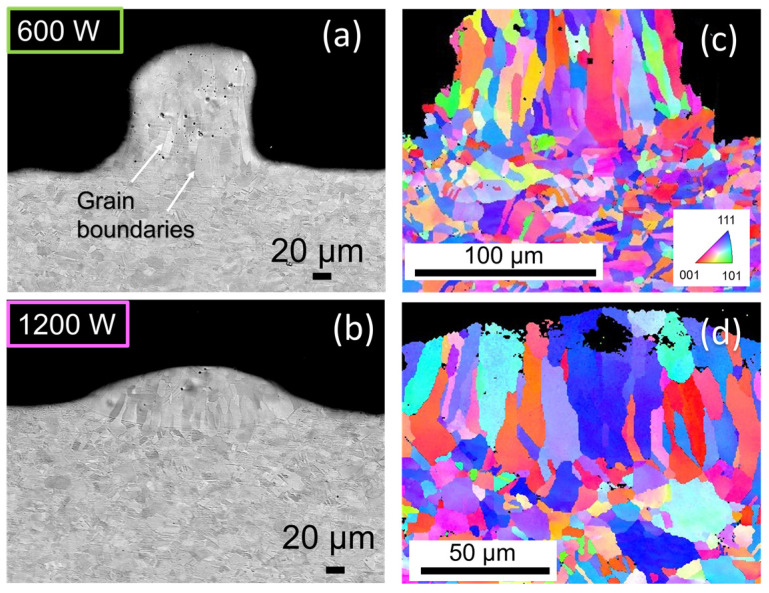
Backscattered SEM images of the samples and inverse pole figure IPF demonstrating the crystallographic orientation of the grains: (**a**,**c**) for C-600 and (**b**,**d**) for C-1200, respectively. The orientation color map is common for (**b**,**d**).

**Figure 7 materials-16-02460-f007:**
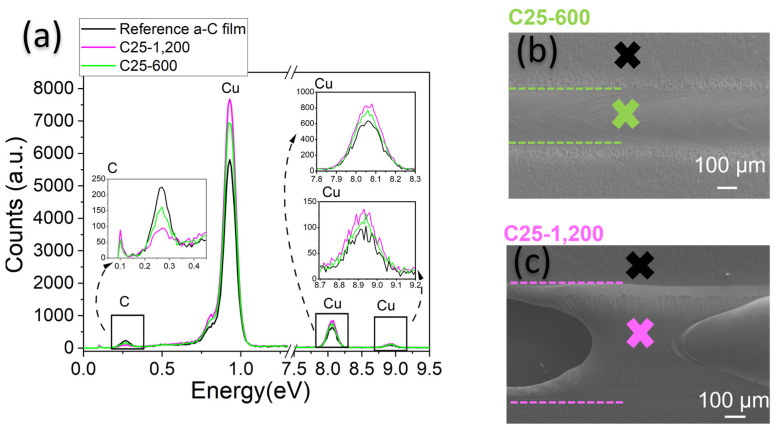
(**a**) Representative EDS of the 25 nm a-C ultra-thin film deposited over a copper foil and the fusion at (**b**) 600 W and (**c**) 1200 W. The color of the crosses in panels (**b**,**c**) indicates the corresponding place of the analysis shown in the spectrum (**a**).

**Figure 8 materials-16-02460-f008:**
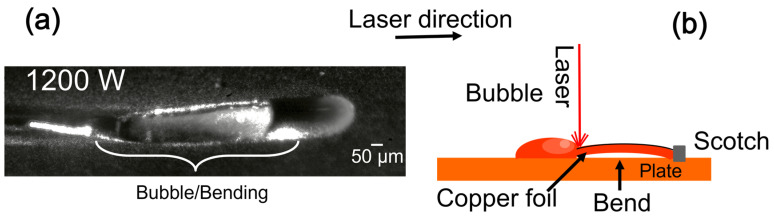
(**a**) Topography recording frame of the fusion using 1200 W during forming a vapor bubble. (**b**) Schematic representation of the cross-section fusion on copper foils.

**Figure 9 materials-16-02460-f009:**
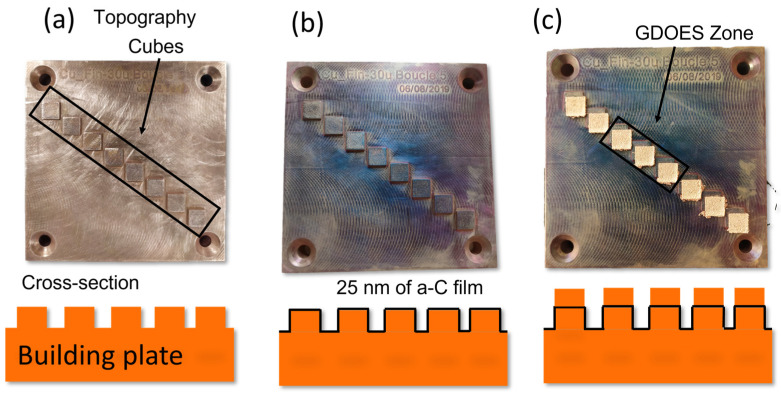
Topography and schematic cross-section representation of (**a**) The squares diagonal printed using SLM, (**b**) Deposition of 25 nm of a-C film by DMCS, and (**c**) Continuation of the printing process over the a-C film and region containing the three cubes analyzed by GDOES (black rectangle).

**Figure 10 materials-16-02460-f010:**
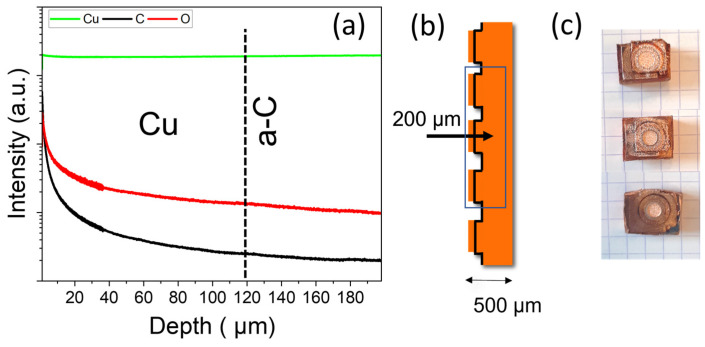
(**a**) GDOES depth analysis presenting the signals of Cu, C, and O emission lines up to 200 µm below the external surface of the sample. (**b**) Schematic representation of the compressed cross-section of the cubes and the building plate. (**c**) Topography of the three analyzed cubes after compression.

**Table 1 materials-16-02460-t001:** Laser main parameters.

Power Range (W)	Power Density (kW/mm^2^)	Spot (µm)	Scan Speed (mm/s)	Focal Length (mm)	Scan Strategy	Powder Temperature	Powder Bed Height (µm)
600–2000	19–64	200 ± 20	500	200	Single tracks	Room temperature	~100

**Table 2 materials-16-02460-t002:** Description of the samples used in this study.

Sample Name	Description
PCu-i	Pure Cu powder
C-i	Cu powder +3.5 µm GRAPHIT 33
FCu-i	Pure Cu foil
a-C25-i	Cu foil +25 nm a-C film
Reference-a-C-film	Cu foil +25 nm a-C
Printed-part-Cu/a-C	Printed cubes +25 nm of a-C

## Data Availability

The interested reader can contact the corresponding authors to obtain research data.

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
