# Peer review of "The Use of Sacrificial Graphite-like Coating to Improve Fusion Efficiency of Copper in Selective Laser Melting"

_materials, 2023, doi:10.3390/ma16062460_

Round 1
Reviewer 1 Report
The manuscript presents a deep analysis into the boosting effect of sacrificial graphite-like coating on the laser absorption of copper metal in SLM. Though the results are interesting and promising while the analysis is very detailed, the experiment design is very confusing. Moreover, the paper is not well prepared with plenty of mistakes. Therefore, I recommend a major revision of the manuscript before publication.
Here are some of the problems:
1. The introduction must be improved to provide more in-depth analysis of the state of the art of the SLM process especially for copper. For example, please provide more data on how much the process has been improved in previous studies. For comparison, the normal processing energy density in previous study should be listed.
2. Please confirm the word ‘selective LAYER melting’ is correct in the title. If this is a mistake, then it is very unacceptable. I strongly recommend the author to go through the whole manuscript and fix similar typos.
3. Based on what result did the author reach the conclusion that the 25 nm a-C coating can boost 40% laser absorption? I did not see any proof or any kind of discussion on how this number is calculated. You need to tell the author and persuade them.
4. Table 2 should include all the experiment parameters. But as far as I see, the experiment for cross-section analysis is not included. (1000 W, 1200 W, 2000 W)
5. Again, if the surface morphology/topography analysis used 600 W and 1200 W for analysis, why not use them in the cross-section analysis? Why use totally new samples? In this way, the readers cannot relay the result of the cross-section analysis to the topography result.
6. More on the parameter choice: two data points (600 W, 1200 W) are really not persuasive for an experimental study. For example, if you want to reach a conclusion that the energy absorption is enough for 1200 W but not for 600 W, then the critical value could be 1199w or 601W. Therefore, I strongly recommend adding more datapoints in the experiment.
7. Figure 6 shows that there is still remaining C element after sintering, but the GDOES result shows none. Can the author have a small discussion on why this is happening? Would it be better if you also proved the EDS result of a none-spattered Cu surface to show that C element is possible from organic contamination?
8. Also, if you tested two points for ref, you should include both results in the figure. Or at least tell the author that there is very small difference between them.
9. Figure 9 did a very poor job in showing where and how the GDOES is conducted. Where is the test point located? On which cube, the center one or the corner one? Why does it not look like a cube from Figure 8 anymore? Please provide the whole image of the printing board.
10. CO2 should be CO2 and there is no space between 2400 and ℃. Please check the whole manuscript for similar mistakes. There should be a space between 100 and μm in figure 6. There should be a multiplication sign between 64 kW and mm -2. In table 1, shouldn’t it be kW/mm2?
11. The method in this manuscript is indeed very interesting. But if the unprinted powders are recycled, as it is a standard procedure in SLM, please comment the viability of the method and add a small discussion paragraph addressing the problem while providing your preliminary solution on solving such problem.
12. If there is no 3.2, please change the section number from 3.1.1, 3.1.2, 3.1.3 to 3.1, 3.2 and 3.3.

Author Response
Dear Reviewer
Thank you very much for all the comments. We carefully considered all your observations.
You can find the answers to your comments in the attached file.
Sincerely
Angela
Sincerely Angela

Reviewer 2 Report
The manuscript entitled “materials-2265001” dealing with laser based powder bed fusion in additive manufacturing has been reviewed. The paper has been nicely written but needs significant improvement. Please follow my comments.
1. What is the main research question for this research work?
2. Why authors selected the laser parameters in Table 1?
3. What is the future direction of this work?
4. Please update the introduction with the new publications in the field. Authors are encouraged to read and add the following two new papers in the field of copper printing.
· Electron beam powder bed fusion of copper components: a review of mechanical properties and research opportunities
5. Please proofread the paper.
6. Laser absorptivity in AM is important which shows the quality of the parts and transition from keyhole to conduction mode. Please read and add the following ref in this area. “The effect of absorption ratio on meltpool features in laser-based powder bed fusion of IN718”.
Author Response
Dear Reviewer
Thank you for your comments. We considered all your observations.
You can find the answers to your comments in the attached file.
Sincerely
Angela

Round 2
Reviewer 1 Report
I recommend to accept this paper in its present form, since it has greatly been improved through revision.
Reviewer 2 Report
The paper is reday to publish.